# Hippo Signaling Pathway in Gliomas

**DOI:** 10.3390/cells10010184

**Published:** 2021-01-18

**Authors:** Konstantin Masliantsev, Lucie Karayan-Tapon, Pierre-Olivier Guichet

**Affiliations:** 1Inserm U1084, Laboratoire de Neurosciences Expérimentales et Cliniques, F-86073 Poitiers, France; konstantin.masliantsev@univ-poitiers.fr (K.M.); lucie.karayan-tapon@chu-poitiers.fr (L.K.-T.); 2Université de Poitiers, F-86073 Poitiers, France; 3CHU de Poitiers, Laboratoire de Cancérologie Biologique, F-86022 Poitiers, France

**Keywords:** gliomas, glioblastomas, hippo signaling pathway, MST1/2, LATS1/2, YAP/TAZ, TEADs

## Abstract

The Hippo signaling pathway is a highly conserved pathway involved in tissue development and regeneration that controls organ size through the regulation of cell proliferation and apoptosis. The core Hippo pathway is composed of a block of kinases, MST1/2 (Mammalian STE20-like protein kinase 1/2) and LATS1/2 (Large tumor suppressor 1/2), which inhibits nuclear translocation of YAP/TAZ (Yes-Associated Protein 1/Transcriptional co-activator with PDZ-binding motif) and its downstream association with the TEAD (TEA domain) family of transcription factors. This pathway was recently shown to be involved in tumorigenesis and metastasis in several cancers such as lung, breast, or colorectal cancers but is still poorly investigated in brain tumors. Gliomas are the most common and the most lethal primary brain tumors representing about 80% of malignant central nervous system neoplasms. Despite intensive clinical protocol, the prognosis for patients remains very poor due to systematic relapse and treatment failure. Growing evidence demonstrating the role of Hippo signaling in cancer biology and the lack of efficient treatments for malignant gliomas support the idea that this pathway could represent a potential target paving the way for alternative therapeutics. Based on recent advances in the Hippo pathway deciphering, the main goal of this review is to highlight the role of this pathway in gliomas by a state-of-the-art synthesis.

## 1. Introduction

The Hippo pathway, initially discovered in *Drosophila*, is a well-conserved evolutionary signaling pathway involved in tissue development and regeneration that controls organ size by regulating cell proliferation and apoptosis [1,2]. Physiologically, the Hippo pathway acts as a tumor suppressor, and the mutations found in different pathway actors induce hyperplasia [3]. The core kinases of the Hippo signaling pathway in mammals is composed of MST1/2 (Mammalian STE20-like protein kinase 1/2) and LATS1/2 (Large tumor suppressor 1/2) corresponding to Hpo (Hippo) and Wts (Warts) in *Drosophila* [4] (Figure 1). The role of these kinases is to regulate negatively the transcription cofactors YAP (Yes-Associated Protein 1) and its ortholog TAZ (Transcriptional co-activator with PDZ-binding motif) corresponding to Yki (Yorkie) in the fly. After receiving activating upstream signals, the pathway is initiated by the activation of MST1/2 associated with SAV1 (Salvador). This complex stimulates by phosphorylation LATS1/2 and its cofactor MOB1, which in turn phosphorylates the transcription cofactors YAP/TAZ, leading to inhibition of its nuclear translocation. In fact, phosphorylated YAP/TAZ is sequestrated in the cytoplasm by 14-3-3 protein. Furthermore, YAP/TAZ stability can be affected via its phosphorylation by the LATS/CK1(ε/δ) (Casein kinase 1 ε/δ) complex, inducing its ubiquitination by the recruitment of E3 ubiquitin ligase SCF-β-TrCP (Skp1-Cul1-F-box *β*-transducin repeat-containing protein), leading to YAP/TAZ degradation by the proteasome. On the other hand, when the Hippo pathway is inactivated, the unphosphorylated YAP/TAZ complex is translocated to the nucleus, where it binds the TEAD (TEA domain) transcription factor family. Indeed, YAP/TAZ association with TEAD transcription factors is essential to the control of several targeted genes such as MYC (MYC proto-oncogene bHLH transcription factor), BIRC5 (baculoviral IAP repeat containing 5), AXL (AXL receptor tyrosine kinase), CTGF (Connective Tissue Growth Factor), or CYR61 (Cysteine Rich Angiogenic Inducer 61) involved in cell proliferation and survival [5].

However, it has also been demonstrated that YAP/TAZ can act in association with other transcription factors, such as RUNX1/2 (RUNT-related transcription factor 1/2) and TBX5 (T-box transcription factor 5) [6,7]. Upstream Hippo pathway activators depend on cellular context. The protein NF2 (Neurofibromine 2), also called MERLIN, represents the most widely described Hippo signaling activator. This cytoskeletal protein is expressed by cell-cell junctions in epithelial cells and stabilizes the interaction between MST1/2 and LATS1/2 [8,9,10]. Other Hippo signaling modulators include DNA damages, contact inhibition, mechano-transduction, and cross-talks with RTKs (Receptor tyrosine kinases), GPCR (G-coupled receptors), WNT (Wingless-type), NOTCH, and SHh (Sonic hedgehog) signaling pathways [3,10,11,12,13].

The first studies investigating the biological role of Hippo signaling were performed using a gain of function approach overexpressing the transcription cofactor YAP. Hippo pathway inactivation by mimicry approach has shown 3-time mass increasing in mice due to uncontrolled cell proliferation [2]. Interestingly, this effect was reversible after stopping YAP overexpression, suggesting the existence of a negative feedback loop between proliferation and apoptosis in Hippo signaling. Other studies have shown the crucial role of MST1/2 and SAV1 in post-natal hepatic growth restriction by maintaining hepatocytes in a quiescent state through YAP inhibition [11,14,15,16]. However, the involvement of the Hippo signaling pathway in organ size regulation is not universal. Indeed, YAP overexpression in small intestine cells has induced tissue hyperplasia and loss of terminal cell differentiation, yet without global organ size increasing in mice [17]. Similar results were obtained after inducing SAV1 loss in epithelial mouse cells [18]. These studies suggested the involvement of Hippo signaling in cell cycle regulation. In fact, Reddy et al. have shown that the Hippo pathway could lead to cell cycle arrest and cell differentiation during the development of the optic neuro-epithelium in *Drosophila* [19]. A key role of YAP has also been shown in eye development in several animal models like zebra-fish and mice, where its inhibition induces a strong retina differentiation defect [20]. Moreover, YAP expression is particularly restricted in mice embryos and some adult tissues. YAP is expressed mainly in the stem or progenitor cells in the skin and intestine, suggesting the involvement of Hippo signaling in cell differentiation and/or stem cell pool maintenance [3]. Indeed, it has been shown that Hippo pathway inhibition increases the pool of progenitor cells rather than differentiated cells in epithelial tissue. In the same way, this pathway is known to modulate stem cell proliferation in *Drosophila* intestine epithelium during its regeneration [21]. Cao et al. have shown that YAP overexpression or MST1/2 inhibition in chicken embryo neuro-epithelium increased the neuronal progenitor cell pool of a spinal cord [22]. In mice, Lian et al. have demonstrated that YAP acts in favor of cell pluripotency by inhibiting embryonic stem cell differentiation [23]. Taken together, these results underlined the crucial role of the Hippo signaling pathway during embryo development and adult tissue homeostasis in the balance between stem cells, progenitor cells and differentiated cells, particularly by controlling cell cycle, apoptosis and cell differentiation processes.

Since MST1/2 and LATS1/2 core constitute a regulatory part of the Hippo signaling associated with a tumor suppressor effect, the transcriptional cofactors YAP/TAZ associated with TEAD transcription factors represent the terminal effectors of this pathway and play a pro-oncogenic role. Recent studies have suggested that YAP and TAZ are essential to the initiation and proliferation of several solid tumors. Indeed, YAP/TAZ activation is involved in cell proliferation, mesenchymal transition, invasion, metastasis formation, as well as in cancer stem cell maintenance and chemoresistance. Constitutive activation of YAP/TAZ is currently known to be associated with aberrant cell proliferation by inducing the expression of several proteins involved in DNA synthesis, replication, reparation, and cell cycle control. For example, YAP/TAZ indirectly enforces cell cycle regulation by inducing other pro-oncogenic transcription factors such as c-MYC [24]. YAP/TAZ likewise contributes to cancer cell survival by the induction of anti-apoptotic proteins of the BCL-2 family, escaping not only mitochondrial apoptosis but also to alternative TNF-α and FAS ligands-induced apoptosis [2,7]. Moreover, YAP/TAZ contributes to anoikis resistance induced by the detachment of cells and substrate [25]. Several studies have shown that YAP/TAZ play a functional role in cancer stem cell maintenance and proliferation [26,27]. Indeed, it was shown that TAZ is crucial for tumor initiation, self-renewal, and metastatic capacity [28,29]. Moreover, YAP/TAZ promotes cancer cell population regeneration by maintaining autophagy basal level to avoid senescence and opposing cell death caused by excessive tumor autophagy [30,31,32]. At the microenvironmental level, YAP/TAZ contributes largely to the interaction between cancer cells and neighboring epithelial cells by inducing the secretion of angiogenic factors such as AREG (an EGF-like growth factor), CYR61, and CTGF. Furthermore, the secretion of chemotaxic molecules induced by YAP/TAZ leads to immune tolerance through suppression of myeloid cells by T lymphocytes [33]. In cancer-associated fibroblasts, YAP and TAZ induce the production of pro-inflammatory interleukins and deposition of a rigid extracellular matrix that is a main upstream inducer of YAP/TAZ, thereby creating positive feedback [34,35]. While all these data support the idea of a large contribution of the Hippo pathway in cancer initiation and progression, the role of the Hippo pathway is still relatively poorly investigated in gliomas.

Gliomas are the most common and the most lethal primary brain tumors and represent about 80% of malignant brain and central nervous system (CNS) tumors [36]. Initially, gliomas were classified according to their histological features, but this classification suffered from high intra/inter-observer variability, which does not sufficiently predict patient outcomes [37,38]. During the past decade, several studies have identified genomic alterations involved in glioma pathogenesis, thereby providing a more accurate stratification than classification based solely on histopathology [39,40]. These key molecular features, including isocitrate dehydrogenase 1 and 2 (IDH1/2) mutations and concurrent loss of both 1p and 19q chromosome arms (1p19q codeletion), have demonstrated their significance in clinical behavior, response to treatment, and patient outcome [41,42]. Indeed, mutations in IDH1/2 and 1p19q codeletion characterize the majority of low-grade gliomas (LGGs) and define a subtype associated with a favorable outcome. On the other hand, glioblastomas (GBMs) (WHO grade IV) are the most common and aggressive form of gliomas [43]. Surgical resection followed by concomitant radiochemotherapy constitutes the gold standard treatment for glioblastoma patients [44]. Despite this intensive clinical protocol, the prognosis for patients remains very poor, with a median survival of 15 months according to tumor invasiveness and radiochemoresistance [45]. Otherwise, treatment failure may also be explained by the persistence of a subpopulation of cancerous cells presenting stem cell capacity termed Glioma Stem Cells (GSCs) [46,47,48]. Since the last decade, the number of studies focused on the role of the Hippo signaling in gliomas is growing but remains relatively poor compared to other solid tumors. The following part of this review is dedicated to the description of the Hippo pathway core actors and their molecular alterations and deregulations, as well as their role reported in the glioma context.

## 2. MST1/2

MST1 is a serine/threonine kinase that plays an important role in organ size regulation by regulating cell apoptosis and proliferation. In response to apoptotic stimuli, MST1 is activated by dimerization-mediated trans-phosphorylation and caspase-mediated cleavage. Cleaved MST1 translocates to the nucleus and induces chromatin condensation by phosphorylation of different targets, but MST1 apoptotic signaling has yet to be completely defined. In 2014, Tang et al. showed that MST1 was negatively regulated by hMOB3 reported to be upregulated in GBM [49]. Indeed, hMOB3 can physically interact with MST1 and prevent MST1, inducing apoptotic signaling. Likewise, Chao et al. showed that MST1 downregulation in U87 and U251 GBM cultures promoted cell growth and proliferation and inhibited apoptosis [50]. Interestingly, MST1 did not affect YAP phosphorylation but was found to bind to AKT and negatively regulate AKT and mTOR activity. MST1 downregulation in glioma could also be induced through TGF-β signaling [51]. Indeed, TGF-β increases DNA methyl-transferase DNMT1 expression, which induces MST1 epigenetic repression in U87 and U251 cells, promoting proliferation, migration, and invasiveness. MST1 expression can also be directly repressed by miR-130b, which can target the MST1-adaptor protein SAV1 [52]. Moreover, miR-130b was found to be upregulated in both GBM tissues and cell lines, which is concordant with MST1 downregulation reported by Zhu et al. [53]. Functional MST1 overexpression has been marked by the induction of SIRT6 (Sirtuin 6), reducing glioma cell viability and colony formation and promoting apoptosis through the activation of FOXO3a transcription factor. Recently, Xu et al. demonstrated that CUL7 can be physically associated with MST1, promoting ubiquitin-mediated MST1 protein degradation, leading to the activation of the NF-κB signaling, a pathway known to be involved in glioma proliferation, migration, and invasion [54].

## 3. LATS1/2

LATS1 and LATS2 are serine/threonine kinases that are direct negative regulators of YAP, playing a pivotal role in organ size control and tumor suppression by restricting proliferation and promoting apoptosis. The human LATS1 gene is localized in chromosome 6q24–25, and its overexpression causes G2-M arrest through the inhibition of CDC2 kinase activity in breast cancer cell lines and significantly suppresses tumorigenicity by inducing apoptosis [55,56,57]. The LATS2 gene is localized on chromosome 13q11–12 and its overexpression has been shown to cause G1-S arrest through the inhibition of cyclin E/CDK2 in vitro and to suppress the tumorigenicity of NIH/v-ras-transformed cells [58,59,60]. In 2006, Jiang et al. showed that the promoter hypermethylation frequencies of LATS1 and LATS2 were 63.66% and 71.5%, respectively, in 88 astrocytomas compared to 10 non-tumoral brain samples presenting an unmethylated promoter profile [61]. LATS1/2 promoter hypermethylation was also found in U251 and SHS-44 GBM cell lines and was associated with correspondingly decreased mRNA expression in astrocytoma samples. These results were confirmed by Ji et al., who found that LATS1 mRNA and protein were significantly downregulated in glioma compared to non-tumoral brain tissues [62]. Interestingly, reduced LATS1 expression was negatively correlated with tumor grade in glioma patients and was associated with significantly shorter overall survival time. Moreover, forced expression of LATS1 in U251 glioma cells not only significantly suppressed cell growth, migration, and invasion but also retarded cell cycle progression from G2/M to G1 in vitro. LATS1/2 activity can also be downregulated indirectly by reducing the protein level of MOB1, an activator of LATS1/2 kinases. Indeed, Lignitto and Arcella et al. showed that the E3-ubiquitin ligase PRAJA2 was able to interact directly with MOB1, inducing its ubiquitylation and degradation by the proteasome [63]. Proteolysis of MOB1 by PRAJA2 attenuates the Hippo cascade and enhances the proliferation of U87MG glioblastoma cells. LATS1/2 degradation can be mediated by IKBKE (inhibitor of nuclear factor kappa-B kinase subunit epsilon), which was shown to be upregulated in glioma [64]. Liu et al. showed that IKBKE knockdown in U87 and U251 GBM cells dramatically elevated LATS1/2 and YAP phosphorylation on S127, suppressing YAP protein and its downstream targets such as AXL, c-MYC, and CYR61 [65]. Furthermore, the authors demonstrated that IKBKE did not alter mRNA levels of LATS1/2 in glioma cells but was directly bound to LATS1/2, and facilitated its polyubiquitin degradation. Recently, Liu et al. highlighted the mechanism by which Ca2+ inhibits YAP/TAZ-mediated transcriptional program through the activation of LATS1/2 in the LN229 glioblastoma cell line [66]. The authors showed that the induced elevation of cytosolic Ca2+ provoked actin cytoskeleton remodeling mediated by INF2, leading to new actin-filament assembly. Ca2+ also induced PKC beta II translocation to the newly formed F-actin compartment, where activated, PKC beta II induced MST1/2 and LATS1/2 phosphorylation, silencing YAP/TAZ transcriptional program. LATS1 can be downregulated by miRNA in glioma. Indeed, bioinformatic analysis has predicted that LATS1 might be a potential target for miR-4262, which has been shown to be upregulated in glioma patient samples compared to normal tissue [67]. In vitro results have suggested that miR-4262 directly and negatively regulates LATS1 expression in U251 cells. Moreover, the authors suggested that overexpression of LATS1 could reverse the effects of miR-4262 suppressing cell proliferation and migration, as well as the production of MMP-2 and MMP-13. Recently, Ji et al. showed that PMEPA1a (prostate transmembrane protein, androgen-induced 1 a isoform) is strongly expressed in human glioma samples and that overexpression increases GBM cell lines growth. Indeed, PMEPA1 promotes LATS1 ubiquitination and degradation by the E3 ligase NEDD4 leading to the inhibition of Hippo pathway and activation of YAP target genes [68]. PMEPA1a was found to be highly expressed in human gliomas, and overexpression of the protein enhanced growth characteristics of glioma cell lines in vitro and in vivo. Although LATS1/2 are considered as tumor suppressors, RASSF1/LATS2-coupled promoter hypermethylation was found to be associated with better overall survival in glioma patients [69]. Hypermethylated promoter profiles were related to IDH mutation, yet not randomly in IDH-mutated gliomas, because LATS2 promoter hypermethylation was more frequent in oligodendroglioma than in astrocytoma.

## 4. YAP/TAZ

Although the role of Hippo signaling pathway in solid tumors is now well established, few studies have investigated its involvement in glioma. It has been shown that TAZ overexpression in GBM is associated with poorer patient prognosis [70,71,72]. Moreover, TAZ inhibition by shRNA in GSC lines reduces its tumorigenicity in SCID (Severe Combined Immunodefeciency) mice. On the other hand, TAZ overexpression in normal central nervous system stem cells and proneural GSCs was sufficient to induce its aberrant osteoblastic and chondrocyte differentiation by inducing mesenchymal transition in association with TEAD2 transcription factor. More recently, Yee et al. confirmed that GBM cell lines expressing a constitutively active form of TAZ induce tumors with mesenchymal features and extensive necrosis [73]. Moreover, the authors showed that necrosis involved neutrophil-triggered ferroptosis in hyperactivated TAZ GBM mouse model. TAZ could also be involved in GBM chemoresistance, as its overexpression in U87 and U251 cell lines was shown to reduce TMZ (Temolozomide) cytotoxicity by induction of MCL-1, leading to apoptosis resistance. Conversely, TAZ inhibition is able to potentialize TMZ effects in GBM lines [71]. Yang et al. have shown that TAZ promotes cell proliferation and tumor formation in U87 and LN229 cells by activating EGFR (Epidermal growth factor receptor) and its downstream AKT and ERK pathways through c-MYC [72]. TAZ can also be involved in radioresistance in GBM. Zhang et al. investigated long-term cellular responses of human GBM cells to ionizing radiation and showed that later response was associated with increased cellular senescence and TAZ inhibition [74]. Mechanistically, TAZ inhibition depends more on increased degradation mediated by the β-catenin destruction complex in the WNT pathway rather than on the canonical Hippo pathway. The authors also showed that TAZ silencing promoted radiation-induced senescence and growth arrest and concluded that TAZ inhibition is implicated in radiation-induced senescence and might improve GBM radiotherapy. TAZ overexpression can be explained by the upregulation of Histone deacetylase 9 (HDAC9) in glioblatoma patients as its knockdown decreases TAZ expression [75]. On the other hand, overexpressed TAZ or TAZ overexpression in HDAC9-knockdown cells abrogated the effects induced by HDAC9 silencing both in vitro and in vivo. Nawaz et al. identified a member of Polycomb Repressive Complex 1 (PRC1) called Chromobox homolog 7 (CBX7) that is downregulated in GBM due to its promoter hypermethylation [76]. Gene set enrichment analysis (GSEA) of CBX7 regulated genes identified CBX7 as a repressor of transcription co-activators YAP/TAZ. Moreover, exogenous expression of CBX7 repressed the YAP/TAZ-dependent transcription and downregulated CTGF, inducing cell death, cell proliferation inhibition, colony formation, migration, and invasion of the glioma cells. Recently, Escoll et al. showed that NRF2 (Nuclear factor (erythroid-derived 2)-like 2) transcription factor related to cellular defense against oxidative stress was involved in tumor progression by providing metabolic adaptation to tumorigenic demands and resistance to chemotherapeutics and that it induces TAZ expression promoting tumorigenesis in GBM [77]. Indeed, expression of the genes encoding NRF2 (NFE2L2) and TAZ (WWTR1) showed a positive correlation in 721 gliomas from TCGA database, which was confirmed by immunohistochemical tissue array analysis at the protein level. Moreover, overexpression and chemical activation or genetic knockdown of NRF2 showed an increase or a decrease of TAZ at both the transcript and protein level in GSC culture. In the same way, studies concerning YAP function have highlighted its involvement in GBM cell line proliferation in vitro and its association with tumor aggressivity [78,79]. Similarly to its ortholog TAZ, YAP in association with TEAD2 favors mesenchymal transition in U87 cells [80]. Moreover, YAP is able to induce WNT signaling transactivation, which plays a major role in GSC maintenance by modulating β-catenin activity through GSK3β regulation [81]. YAP also seems to closely interact with the PI3K/AKT/mTOR (Phosphatidylinositol 3-kinase/AKT serine/threonine kinase/Mechanistic target of rapamycin kinase) signaling pathway in glioma. Liu et al. have shown a positive correlation between the phosphorylated form of mTOR and unphosphorylated YAP protein expression [78]. Moreover, the combination of p-mTOR and YAP expression was negatively related to the overall survival of patients and associated with a high grade of glioma. In the same way, Artinian et al. reported that YAP activation in glioma cell lines could be supported via mTOR through the inhibition of AMOTL2 (angiomotin-like 2) [82]. The angiomotin family members directly interact with YAP and the actin cytoskeleton, promoting YAP inactivation by cytoplasmic retention and leading to YAP phosphorylation by Hippo signaling [83]. Moreover, the authors showed the capacity of mTORC2 to inhibit AMOTL2 activity post-translationally by phosphorylation leading to increased YAP function. Indeed, this regulatory phosphorylation prevents AMOTL2 binding to YAP and stimulates YAP transcriptional program associated with enhanced growth and invasiveness in glioma cell lines. Liu et al. demonstrated that YAP1 expression participates to intratumoral heterogeneity in GBM [84]. Indeed, strong expression of YAP1 promotes tumorigenesis and clonal dominance accompanied by growth enhancement. Moreover, the authors suggested that cellular interaction during clonal dominance induces tumorigenic gene expression contributing to tumor growth. YAP/TAZ can be stabilized by actin-like 6A (ACTL6A), which has been shown to be upregulated in glioma and associated with patient survival [85]. Ji et al. have shown by co-immunoprecipitation assays that ACTL6A can physically associate with YAP/TAZ and disrupts the interaction between YAP and SCF-β-TrCP E3 ubiquitin ligase, preventing YAP protein degradation. Strong YAP expression was found to be associated with aggressive glioma molecular subtypes, i.e., IDHwt gliomas, as well as overall patient survival and progression-free survival [86]. More interestingly, YAP could be considered as an independent prognostic factor in lower-grade gliomas. Otherwise, at a cellular level, YAP could play a role in GSC proliferation and phenotype maintenance, notably by repressing the OLIG2 (Oligodendrocyte transcription factor 2) proneural factor. Vigneswaran et al. also showed that YAP/TAZ transcription cofactors regulate the expression of SOX2 (SRY-box transcription factor 2), C-MYC, and EGFR to create a feedforward loop to maintain the proliferation and survival of EGFR-amplified/mutant human GBM cells [87]. Another recent study demonstrated that YAP/TAZ are required for the oncogene-dependent transformation of primary neural cells, maintain GSC phenotype, prevent GSC differentiation, and control GBM cell plasticity showing the pivotal role of these transcriptional cofactors in glioma pathogenesis [88].

## 5. TEADs

Little is currently known concerning the role of the TEAD transcription factor family in glioma. In 2011, Bhat et al. showed that TEAD2 was involved mainly in glioblastoma mesenchymal transition [70]. Using chromatin immunoprecipitation, the authors showed that TEAD2, associated with TAZ, binds to a majority of mesenchymal gene promoters. In a murine model of glioma, the coexpression of TAZ, but not a mutated form of TAZ that lacks the TEAD binding site, with platelet-derived growth factor-B (PDGF-B) induced high-grade tumors with mesenchymal traits. The result was confirmed by Lu et al., showing that TEAD2 in association with YAP favored mesenchymal transition in U87 cells [80]. TEAD1 was found to contribute to EGFR effect amplification by inducing c-MYC expression, which fixes and activates EGFR promoter in U87 cells [89]. TEAD4 was shown to interact directly with TAZ and could be involved in cell proliferation, migration, and invasion as well as in mesenchymal transition in glioma cells [90]. In 2018, Tome-Garcia et al. performed a comparative analysis of chromatin accessibility using an assay for transposase accessible chromatin with sequencing (ATAC-seq) to highlight the differences between neural stem/progenitors and GSCs [91]. The authors identified the transcriptionally accessible regions that are specifically related to GSC migration and enriched for TEAD1/4 motifs. TEAD1 knockout by the CRISPR-Cas9 gene-editing technique showed a decrease of mesenchymal transition genes and cell migration in vitro and in vivo. Moreover, the authors showed that TEAD1 directly regulates Aquaporin 4 (AQP4) and that the overexpression of both TEAD1 and AQP4 is sufficient to restore migratory defects in TEAD1-KO cells.

## 6. Hippo Signaling Pathway Signatures

Recently, Wang et al. deciphered the molecular alterations and regulations of 19 Hippo core genes in 9125 tumor samples, using multidimensional omic data from the TCGA [92]. The authors developed a YAP/TAZ transcriptional target signature of 22 genes, which has shown strong prognostic significance through several cancer types. More particularly, the expression of YAP and TAZ at the mRNA and protein level and the expression of YAP/TAZ target genes showed a significant association with worse survival in lower-grade gliomas. Futhermore, the YAP/TAZ target signature was positively correlated with tumor-infiltrating immune cell abundance: Macrophages, CD4 T-cells, neutrophil, and dendritic cells. Finally, the combined analyses of mutations, somatic copy number alterations, methylation, and expression data in lower-grade gliomas showed that the main alterations of Hippo core genes affected YAP/TAZ and TEAD3-4 principally at the expression level. More recently, Kim et al. evaluated the clinical significance of Hippo signaling in glioblastoma by generating a core gene expression signature from four different previously established silences of Hippo pathway (SOH) signatures in the stomach, liver, ovarian, and colorectal cancers [93]. A SOH and active Hippo pathway (AH) from the Hippo core gene expression signature was predicted and validated in glioblastoma samples from The Cancer Genome Atlas (TCGA) and the study of Gravendeel et al. [94]. In both GBM cohorts, the SOH signature was associated with lower overall survival compared to the AH signature. Gene expression and network analysis revealed that SOH subgroup is particularly enriched in genes linked to immune response, mesenchymal transition, and YAP transcriptional targets such as CTGF and CYR61. Interestingly, inhibitory immune checkpoint and M2-polarized macrophages genes were increased in the SOH group suggesting GBM resistance to host immune response. Finally, the SOH signature was strongly correlated with a bad prognosis of GBM patients and may be mediated by pro-tumoral immunosuppression. Taken together, these data show an association between the YAP/TAZ/TEADs Hippo pathway effectors and glioma patient outcome.

## 7. Therapeutic Perspectives

All the data presented above indicate the involvement of Hippo signaling in glioma pathophysiology and support the development of therapeutic avenues targeting this pathway (Figure 2). Despite intensive clinical protocols in the management of glioblastoma, i.e., surgery associated with concomitant radiochemotherapy, no curative treatments exist to date. Indeed, relapse ineluctably occurs, principally due to local invasiveness, therapeutic failure in radiation therapy, and drug response, as well as the existence of cancer stem cell population. In the past, several studies and clinical trials have investigated the potential of signaling-targeted therapy against EGFR, PDGFRA (Platelet derived growth factor receptor alpha), FGFR (Fibroblast growth factor receptor), VEGF (Vascular endothelial growth factor), STAT3 (Signal transducer and activator of transcription 3), and PI3K without significant improvement in overall patient survival. Increased evidence of Hippo pathway involvement in tumor progression and resistance to treatment has led to the recent development of specific inhibitors targeting YAP/TAZ-TEAD signal transduction. Currently, more than 50 drugs have been shown to inhibit YAP/TAZ activity. However, with the exception of verteporfin, none act directly on YAP/TAZ [95]. There are three main approaches to target the Hippo pathway (i) stimulating the LATS1/2-dependent inhibitory phosphorylation of YAP/TAZ (ii) directly inhibiting the TEAD transcription factors or disrupting the formation of the YAP/TAZ-TEAD complex, and (iii) targeting oncogenic proteins that are transcriptionally upregulated by Hippo effectors. Although stimulating the LATS1/2 kinases by upstream signals showed YAP/TAZ inhibition, these drugs are not specific and do not take into account existing signaling cross-talks leading to YAP/TAZ activation independently from the canonical Hippo pathway.

On the other hand, YAP/TAZ-TEAD downstream oncogenic proteins appear difficult to target due to the number of regulated genes and their activation by other pathways [24,92]. Thus, the most efficient strategy to inhibit the hippo signaling pathway effectors seems to directly target YAP/TAZ-TEAD interaction. One of the first and most popular Hippo pathway inhibitors developed is a benzoporphyrin molecule called Verteporfin. This small inhibitor can directly bind YAP, disrupting its interaction with TEAD transcription factors [96]. However, the exact mechanism by which Verteporfin binds to YAP remains poorly understood, and some inhibitory effects appear to be YAP-independent [97,98]. Verteporfin induced apoptosis and suppressed expression of YAP/TAZ transcriptional targets of patient-derived EGFR-amplified/mutant GBM cells leading to better survival in an orthotopic xenograft model [87]. More interestingly, the administration of Visudyne, an FDA-approved form of Verteporfin, in patients with suspected or known recurrent GBM showed its absorption in GBM tumor cells, suggesting the use of Verteporfin as a promising therapeutic agent for EGFR-amplified/mutant GBM. Recently, a VGLL4-mimicking peptide, also known as Super-TDU, showed disruption of YAP-TEAD interaction and may represent a promising new therapeutic strategy against YAP-driven human cancers [99]. The VGLL family represents another TEAD co-regulator protein, which acts as a competitor of YAP and TAZ [5,100]. More recently, a fused tricyclic compound called CA3, and new molecules targeting hydrophobic pocket of TEADs identified by virtual screening approach were shown to inhibit YAP/TEADs transcriptional activity [101,102]. Growing evidence demonstrating the role of Hippo signaling in cancer biology and the lack of efficient treatments for glioblastoma support the idea that Hippo effectors YAP/TAZ-TEADs could represent potential targets paving the way for alternative therapeutics. Thus, it seems essential that the use of vectorized targeted drugs against the Hippo pathway alone or in combination with conventional GBM therapies, i.e., radiotherapy and Temozolomide, to assess their potentiation or synergistic effects should be proposed in the near future.

## Figures and Tables

**Figure 1 cells-10-00184-f001:**
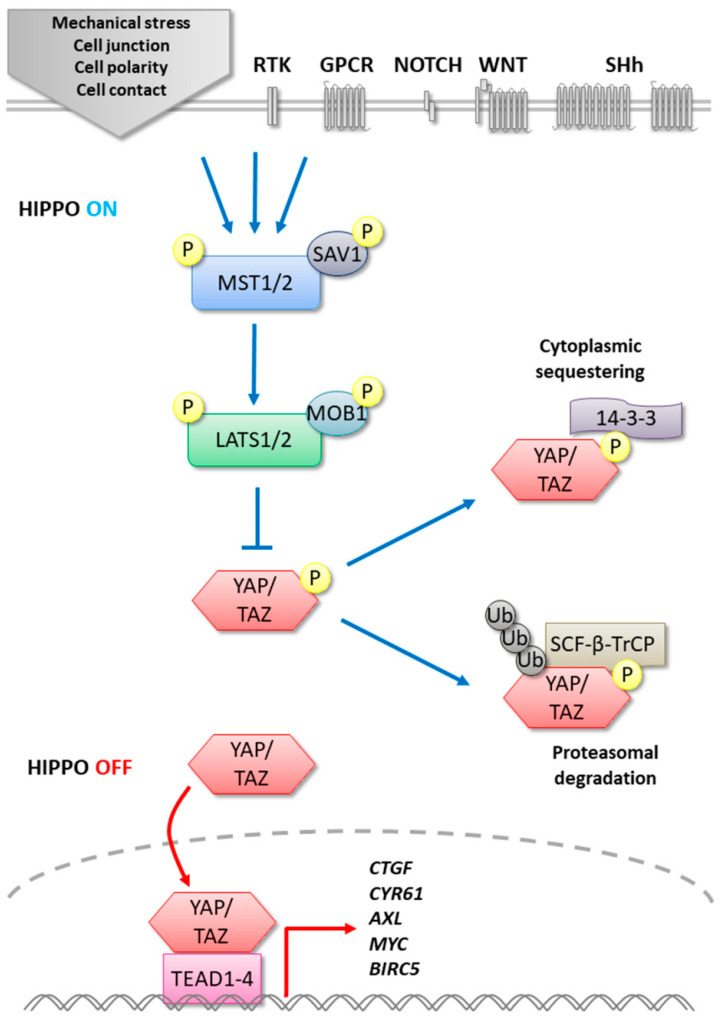
Canonical Hippo signaling pathway in mammals.

**Figure 2 cells-10-00184-f002:**
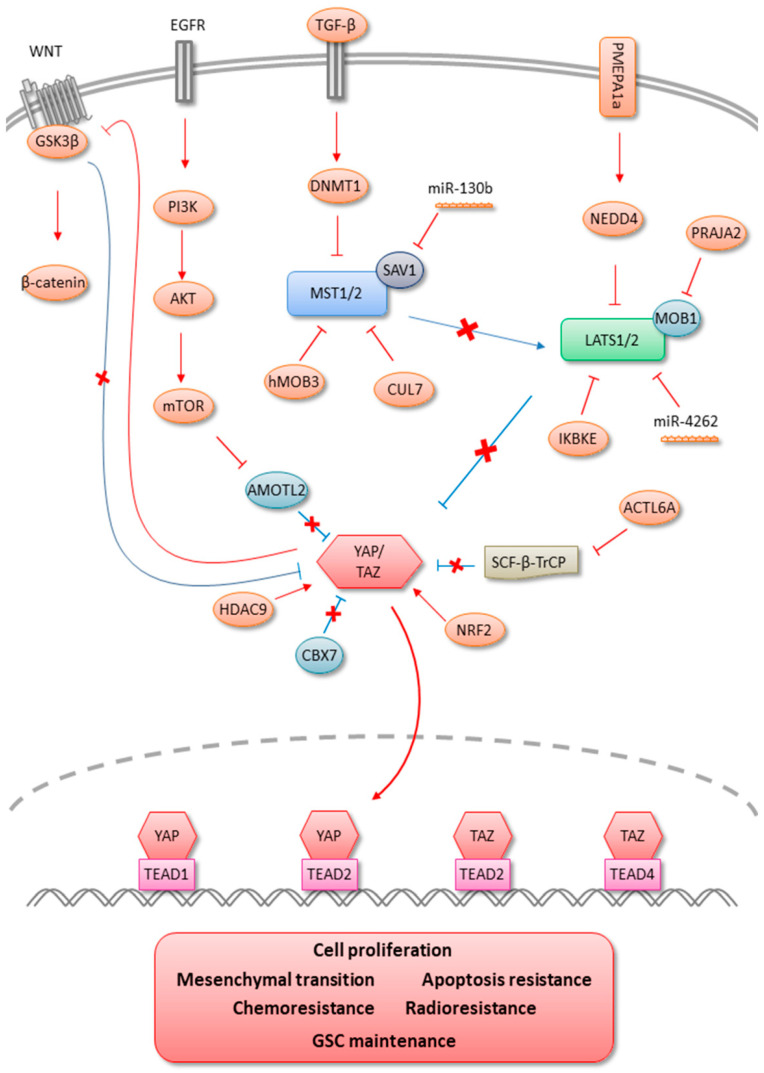
Modulations of Hippo signaling in gliomas.

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
