# Peer review of "Hippo Signaling Pathway in Gliomas"

_cells, 2021, doi:10.3390/cells10010184_

Round 1
Reviewer 1 Report
The manuscript # 1059608 submitted to your journal entitled: “Hippo Signaling Pathway in Gliomas” by Konstantin Masliantsev, Lucie Karayan-Tapon and Pierre-Olivier Guichet is a very concise and well written review on a subject which is very well documented.
It includes two diagrams, one depicting the Hippo signaling pathway in mammals and a second one with the modulation of this pathway in gliomas.
Although it does include 90 references, among the numerous reviews that are available on the web there are the following:
Meng Z, Moroishi T, Guan KL. Mechanisms of Hippo pathway regulation. Genes Dev. 2016 Jan 1;30(1):1-17. doi: 10.1101/gad.274027.115. PMID: 26728553; PMCID: PMC4701972.
Yu FX, Guan KL. The Hippo pathway: regulators and regulations. Genes Dev. 2013;27(4):355-371. doi:10.1101/gad.210773.112
that are not included. More publications specific to tumorigenesis:
Barron, D.A., Kagey, J.D. The role of the Hippo pathway in human disease and tumorigenesis. Clin Trans Med 3, 25 (2014). https://doi.org/10.1186/2001-1326-3-25
and even more specific to glioma:
van den Bent MJ, Mellinghoff IK, Bindra RS. Gray Areas in the Gray Matter: IDH1/2 Mutations in Glioma. Am Soc Clin Oncol Educ Book. 2020 Mar;40:1-8. doi: 10.1200/EDBK_280967. PMID: 32186930; PMCID: PMC7673204.
Venneti S, Huse JT. The evolving molecular genetics of low-grade glioma. Adv Anat Pathol. 2015 Mar;22(2):94-101. doi: 10.1097/PAP.0000000000000049. Erratum in: Adv Anat Pathol. 2015 May;22(3):226. PMID: 25664944; PMCID: PMC4667550.
especially the latest one:
Ouyang T, Meng W, Li M, Hong T, Zhang N. Recent Advances of the Hippo/YAP Signaling Pathway in Brain Development and Glioma. Cell Mol Neurobiol. 2020 May;40(4):495-510. doi: 10.1007/s10571-019-00762-9. Epub 2019 Nov 25. PMID: 31768921.
are also not included to the references of the manuscript.
Additionally, the conclusion of the manuscript states that: …. the lack of efficient treatments for glioblastoma support the idea that Hippo effectors YAP/TAZ-TEADs could represent potential targets paving the way for alternative therapeutics.
A search of the “Hippo effectors YAP/TAZ-TEADs and glioma” had shown over 15 results relevant to the subject.
:https://scholar.google.gr/scholar?q=Hippo+effectors+YAP/TAZ-TEADs+and+glioma&hl=en&as_sdt=0&as_vis=1&oi=scholart
The fact that all existing references are not included is expected since Hippo signaling is very well documented and there is a limitation on the pages in the publication to be submitted.
There is also a diagram in the manuscript which is devoted to the possible modification of the Hippo pathway in gliomas adding information to many existing publications on the subject.
In conclusion, it is really an excellently written review on a very well documented issue from knowledgeable scientists.
But I am not persuaded that the review has added something new and worthy being published by your journal on the Hippo signaling in gliomas. Especially the recent publication of Quyang et al., is a thorough review on brain development and gliomas devoting paragraphs to: The Hippo/YAP Signaling Pathway in Glioma and even more in MST1/2, LATS1/2 and YAP in Glioma suggesting at the end a Therapeutic Prospect of Hippo/YAP Signaling in Glioma.
Minor detail: Drosophila needs to be in italics
Author Response
We would like to thank the reviewer for his comments to improve our manuscript. We agree that the literature concerning the Hippo pathway is substantial and that it is difficult to include all references. However, we inserted all the references proposed by the reviewers and a very recent one, which adds new interesting data (Castellan et al., 2020) concerning the major role of Hippo signaling in gliomagenesis.
Since the last decade, the investigations of this pathway in different solid tumors are in full development but 15 results in gliomas after querying the google scholar database is few compared to the most common alterations in gliomas such as the IDH1 mutation or EGFR amplifications/mutations showing more than 20,000 and 100,000 results respectively.
To be honest with the reviewer, we received an invitation to submit a paper for the special issue of Cells journal. As our main team project is currently focused on the Hippo pathway in glioblastoma and GSCs, we proposed a review dedicated to this signaling. We strongly believe that our manuscript is complementary to that of Ouyang et al. as it deals with MST1/2, LATS1/2 and YAP, but also TAZ, TEAD transcription factors and recent Hippo signatures established from omics data. For that reason, we are convinced that this review could be of general interest for scientists and clinicians working in the fields of Hippo pathway, gliomas and others cancers.
Reviewer 2 Report
Although the Hippo pathway has been shown to be involved in multiple cancers, it is still poorly studied in brain tumors. Recently, some studies have suggested that this conserved tumor suppressor pathway also plays important roles in these tumors, including gliomas. In this manuscript, Masliantsev et al., reviewed recent progress in the research of the regulation and function of this pathway in gliomas, as well as the potential therapeutics based on the knowledge. The manuscript is well organized and should be of interest to researchers in the fields studying the Hippo pathway, glioma and other cancers.
One issue that the authors may consider to add before publication is what are the important questions or problems which are to be answered or resolved currently for this topic. In addition, there are several recent publications related to the topic in the manuscript should be discussed. These include,
- Liu, 2019, Journal of Cell Science. Differential YAP expression in glioma cells induces cell competition and promotes tumorigenesis.
- Yee, 2020, Nature Communications. Neutrophil-induced ferroptosis promotes tumor necrosis in glioblastoma progression.
- Vigneswaran, 2020, Clinical Cancer Research. YAP/TAZ transcriptional co-activators create therapeutic vulnerability to verteporfin in EGFR mutant glioblastoma.
Author Response
We thank the reviewer for the careful review of our manuscript. The comment concerning the questions and problems that have to be answered or solved is perfectly right. Indeed, the number of questions remains numerous especially concerning the exact role of Hippo signaling in tumorigenesis but will be probably solved in the near future. A very recent (December 2020) and elegant publication from Castellan et al. in Nature Cancer showing the pivotal role of the hippo pathway in gliomagenesis solved a part of those problems. We added some sentences regarding this publication in our manuscript. We also included the three articles proposed by the reviewer. We found the publication from Vigneswaran et al. particularly interesting asking us for the best way to treat GBM with specific and vectorized hippo inhibitors. We strongly think that the combination effect of Visudyne or other encapsulated targeted drugs against the Hippo pathway on conventional GBM therapies should be assessed as it was demonstrated for other targeted drugs. We also added a sentence in the manuscript to address this point.
Round 2
Reviewer 1 Report
Having reviewed the modifications/additions and explanations the authors have made into the manuscript I suggest its publication to your journal.